# Strategies for Bottlenecks of rAAV-Mediated Expression in Skeletal and Cardiac Muscle of Duchenne Muscular Dystrophy

**DOI:** 10.3390/genes13112021

**Published:** 2022-11-03

**Authors:** Na Li, Yafeng Song

**Affiliations:** 1School of Sports Science, Beijing Sport University, Beijing 100084, China; 2China Institute of Sport and Health Science, Beijing Sport University, Beijing 100084, China

**Keywords:** Duchenne muscular dystrophy, gene therapy, MyoAAV, CDCs, saRNA, 5′UTR

## Abstract

Gene therapy using the adeno-associated virus (rAAV) to deliver mini/micro- dystrophin is the current promising strategy for Duchenne Muscular Dystrophy (DMD). However, the further transformation of this strategy still faces many “bottlenecks”. Most gene therapies are only suitable for infants with strong muscle cell regeneration and immature immune system, and the treatment depends heavily on the high dose of rAAV. However, high-dose rAAV inevitably causes side effects such as immune response and acute liver toxicity. Therefore, how to reduce the degree of fibrosis and excessive immune response in older patients and uncouple the dependence association between therapeutic effect and high dose rAAV are crucial steps for the transformation of rAAV-based gene therapy. The article analyzes the latest research and finds that the application of utrophin, the homologous protein of dystrophin, could avoid the immune response associated with dystrophin, and the exploration of methods to improve the expression level of mini/micro-utrophin in striated muscle, combined with the novel MyoAAV with high efficiency and specific infection of striated muscle, is expected to achieve the same therapeutic efficacy under the condition of reducing the dose of rAAV. Furthermore, the delivery of allogeneic cardio sphere-derived cells (CDCs) with anti-inflammatory and anti-fibrotic characteristics combined with immune suppression can provide a continuous and appropriate “window period” for gene therapy. This strategy can expand the number of patients who could benefit from gene therapy.

## 1. Introduction

Duchenne muscular dystrophy (DMD) is an X-linked recessive neuromuscular disorders, with a prevalence rate of 1/3500–5000 newborn boys worldwide [1]. The main symptoms of DMD are progressive muscle weakness accompanied by muscle steatosis. Affected boys usually present clinical symptoms between 3 and 5 years old, they lose ambulation ability around 12 years old and die in the second or third decade due to cardio-respiratory insufficiency [2].

DMD is caused by a mutation in the *DMD* gene that encodes dystrophin, which is the largest known human gene, comprising 79 exons and spanning 2600 kb, roughly 0.1% of the whole genome. More than 7000 mutations were found in affected boys, including deletion spanning one or multiple exons, which account for 60 to 70% of all DMD cases. The point mutations are responsible for 26% of DMD cases. Exonic duplications account for 10% to 15% of DMD cases [3]. Dystrophin is located under the sarcolemma and forms the dystrophin-associated glycoprotein complexes (DGC) with a group of transmembrane proteins. The DGC complexes form a mechanical connection between the intracellular skeleton and the extracellular matrix, which together protect the sarcolemma from the damage caused by the stress generated during muscle contraction [4,5]. In the absence of dystrophin, the sarcolemma is unable to recruit DGC complexes, and thereby becomes extremely unstable, resulting in muscle fiber necrosis, inflammatory cell infiltration, muscle fiber regeneration, and eventually muscle fiber replacement by connective tissue with progressive muscle sclerosis and loss of contractile function. With the disease continuing to advance, the regeneration ability of muscle fibers decreases and gradually degenerates and disappears, finally involving respiratory muscles and myocardium [6,7].

Since the identification of the *DMD* gene in 1986, the search for a cure for DMD has been intense. However, there is currently no effective form of a cure because the significant obstacles include both the large size of the *DMD* gene for gene therapy and that affecting nearly all muscle types (skeletal, smooth and cardiac muscle, which together represent ~40% of the total body mass in males) as well as the limited “treatment window”. Comprehensive analysis found that in addition to health management, drugs targeting different *DMD* mutation types have been developed, but these drugs are only suitable for patients with specific mutation types, and their clinical benefits are very limited. Many studies have demonstrated that gene therapy based on the recombinant adeno-associated virus (rAAV) to deliver mini/micro- dystrophin is the best strategy for the treatment of DMD. However, preclinical and clinical trials have demonstrated that therapeutic efficacy depends on patient age and disease deterioration, and high vector titers that can achieve significant therapeutic efficacy can cause severe liver toxicity. Therefore, maintaining and extending the “window period” of gene therapy and reducing the coupling degree of vector titer and liver toxicity are important bases to ensure the continuous and stable effect of gene therapy, and are strong guarantees to benefit more DMD patients in the whole life cycle.

## 2. Current Situation of DMD Treatment

Clinical multidisciplinary disease management is mainly based on glucocorticoid therapy, scoliosis surgery, respiratory assistance and physical therapy. These methods can protect muscle tissue and function, improve the quality of life and extend the life span of patients to a certain extent, but they cannot change the degenerative process of DMD. In recent years, in addition to comprehensive health management, a variety of emerging strategies have been used to treat DMD and achieved remarkable results. Treatment strategies currently in preclinical and clinical trials are focused on two broad categories: (1) small molecule drugs targeting *DMD* gene mutations aimed at restoring endogenous dystrophin reading frames, and (2) rAAV vector-based gene therapies.

In the clinical treatment, glucocorticoids play a major role in slowing down the speed of skeletal muscle atrophy by regulating the proportion of T lymphocyte subsets and inhibiting the excessive cellular immune response. However, long-term use of glucocorticoids can cause considerable side effects, including decreased bone mineral density, weight gain, height suppression, impaired glucose tolerance, and behavioral changes [8]. Scoliosis surgery, respiratory assistance, and physical therapy are used to improve symptoms of muscle weakness in patients with varying degrees of severity. In addition, angiotensin converting enzyme inhibitors and β-adrenoceptors are commonly used to treat the cardiomyopathy caused by DMD. Bisphosphonates are used to treat muscle weakness, decreased mobility and other related symptoms [9,10]. Nonsense codon readthrough and exon skipping are widely investigated strategies aiming to restore endogenous dystrophin expression. Nonsense codon readthrough utilizes certain medications to selectively induce the ribosomal read-through of premature stop codons without affecting normal termination. Exon skipping uses oligonucleotide backbone chemistries to target the single-exon and multi-exon deletions in the dystrophin gene, which are the most common causes for DMD. Antisense oligonucleotides (AON) are being employed to modify post-transcriptional RNA, resulting in one or multiply exon deletions that restore the transcript back into the normal reading frame and enable a truncated dystrophin. PTC124 (also named as ataluren), a candidate with the capacity to promote UGA nonsense suppression with low toxicity, has been widely investigated in animal models and DMD patients [11]. However, in a phase III trial (NCT01826487), the change in a 6-min walking distance (6MWD) did not differ significantly between patients in the ataluren group and those in the placebo group and thereby was rejected by the US Food and Drug Administration (FDA) in 2016 [12]. The exon skipping AON targeting exon 51 (eteplirsen) [13,14], exon 53 (golodirsen, viltolarsen) [15,16], and exon 45 (casimersen) [12] has been approved by FDA. However, the applicable patient population and therapeutic efficiency remain limited. Those AONs exhibited a lack of efficacy, and the increased dystrophin production in DMD patients was less than 15% of the normal levels, which was conferred to have substantial therapeutic benefits [17]. Moreover, the effectiveness has not been fully confirmed in clinical practice. The accelerated approval of exon skipping drugs was based on the urgent needs of patients with severe neurological diseases in addition to the preliminary clinical data, more clinical benefits of this therapy are expected to be learned from ongoing confirmatory clinical trials.

Gene therapies, strategies that rely on the production of recombinant genetic materials, have the potential to provide long-lasting therapy with a single treatment for some diseases. For DMD, rAAV-mediated mini/micro-dystrophin for gene replacement and Crispr/Cas9 for gene editing have been extensively investigated to develop a cure. AAV9-mediated four non-coding U7 small nuclear RNA (AAV9.U7snRNA U7snRNA) targeting the splice donor, and acceptor sites of dystrophin exon 2 have been demonstrated to induce the skipping of exon 2 and expression of the full-length dystrophin in the *Dmd* exon 2 duplication (Dup2) mouse model. Furthermore, low off-target activity and the lack of toxicity were determined after local splice variation (LSV) analysis in mice and toxicity tests in nonhuman primates receiving clinically relevant doses of an AAV9.U7snRNA vector [18,19]. The robust results promoted the phase-I/II clinical trial for scAAV9.U7.ACCA in early 2020 (NCT04240314), which will provide critical information about the safety and efficacy of scAAV9.U7.ACCA by 2025 [20]. However, there have been no promising findings for other exon skipping therapies. Crispr/Cas9-based gene editing approaches to restore the *DMD* open reading frame and rescue the functional dystrophin expression have recently been demonstrated in large animal models of DMD [21]. Co-delivered AAV9-Cas9 and AAV9-gRNA targeting the *DMD* gene 51 splice acceptor site have been demonstrated to rescue dystrophin of up to 70% and 92% of the wild-type dystrophin levels, 8 weeks post-treatment in the peripheral and cardiac muscles of deltaE50-MD dogs, respectively [22]. Moreover, dual AAV9s could be used to co-deliver a split-intein Cas9 and a pair of gRNAs to excise exon 51 and restore the reading frame in *DMD* exon 52-deficient pigs [23,24]. However, CRISPR technology could offer a one-time treatment for muscular dystrophies by correcting the culprit genomic mutations and enabling a normal expression of the repaired gene, it still belongings to the mutation specific therapy, further studies were needed to assess the off-target and site-specific editing efficiency. Comparing to the mutation specific therapy (such as nonsense codon readthrough, exon skipping and rAAV-mediated gene editing), rAAV-mediated gene replacement therapy, which is irrespective of the mutation type, is potentially an improved strategy for targeting a broader cohort of DMD patients.

## 3. Clinical Development of Systemic AAV-Mediated Mini/Micro-Dystrophin Gene Therapy

rAAV, derived from the wild-type AAVs, are well characterized as potential gene transfer vehicles in the treatment of neuromuscular disorders, with the capacity to transduce the vast majority of the striated musculature with a single administration. AAV is composed of an icosahedral protein capsid of ~26 nm in diameter and a single-stranded DNA genome of ~4.7 kb that can either be the sense or anti-sense strand. The capsid comprises three types of subunit, VP1, VP2 and VP3, in a ratio of 1:1:10 (VP1:VP2:VP3). The genome is comprised of three genes (*Rep*, *Cap* and *aap*), which are flanked by two T-shaped inverted terminal repeats (ITRs) that largely serve as the viral origins of replication and the packaging signal [25]. The *Rep* encodes proteins required for gene replication, *Cap* gene encodes the three capsid subunits and recognizes the host cell surface and the *aap* gene encodes the assembly-activating protein needed for the capsid assembly and nuclear localization for some AAV serotypes [26]. However, rAAV genomes substitute all the AAV protein-coding sequence with therapeutic gene expression cassettes, and just reserve the ITRs, which are needed for gene replication and packaging during vector production. The complete removal of viral coding sequences maximizes the packaging capacity of rAAVs and contributes to their low immunogenicity and cytotoxicity when delivered in vivo [27].

To circumvent the difficulty in AAV packaging with the full-length dystrophin gene, mini/micro-dystrophin therapies has been developed. The notion of miniaturized dystrophin was based on a Becker muscular dystrophy patient, who remained ambulatory for seven decades despite a deletion of nearly half of the *DMD* gene [28]. Recently, mini-dystrophin incorporating different spectrin repeats (SRs) and hinges (Figure 1) delivered by different AAV serotypes are undergoing assessment of safety and efficacy in three simultaneous gene therapy trials (Table 1) [29] sponsored by Sarepta Therapeutics, Pfizer, and Solid Biosciences.

### 3.1. Sarepta-SRP-9001

Sarepta Therapeutics used rAAVrh74 (a serotype very similar to AAV-8) to delivery of the micro-dystrophin driven by MHCK7 promoter [30], named as SRP-9001. MHCK7 promoter (~770 bp), a novel regulatory cassette based on enhancer/promoter regions of murine muscle creatine kinase (CK) and α-myosin heavy-chain genes, can direct high-level expression comparable to cytomegalovirus and Rous sarcoma virus promoters in fast and slow skeletal and cardiac muscle, and low expression in the liver, lung, and spleen [30]. According to the phase 1/2 open-label, safety, and tolerability trial (ClinicalTrials.gov: NCT03375164), 4 DMD boys, mean age 4.8, with mutations between exons 18 and 58, took prednisolone for ≥12 weeks but the AAVrh74 total binding antibody titers < 1:400 were enrolled [31]. The subjects received a daily dose of 1 mg/kg prednisone for 30 days before the 2.0 × 10^14^ vg/kg SRP-9001 was injected. After 12 weeks, therapeutic efficiency was assessed. Compared to the baseline, the gastrocnemius muscle biopsy showed a mean of 81.2% muscle fiber expression of micro-dystrophin, and the mean expression level was 74.3%. Serum CK remained decreased in all subjects (range 46~85%), with functional improvement by a mean 5.5 points in the North Star Ambulatory Assessment (NSAA) at 1 year. Meanwhile, the adverse event (AE) profile was minimal. The transient elevation of γ-glutamyltransferase in three patients was resolved with corticosteroids and the most common was vomiting (50%), which was considered to be not correlated with the AAV immunity. Currently, the treated patients average over 9 years old and in the predicted steep decline phase of disease they did not decline, but demonstrated a 7-point increase above their pre-treatment baselines on NSAA, and a 9.9-point (unadjusted means) and 9.4-point (least squared means) improvement versus a propensity-weighted external control (*p* = 0.0125) [32].

NCT03769116 is a randomized double-blind placebo-controlled phase 2 trial assessing the efficacy of SRP-9001 in 41 males with DMD. DMD patients between the ages of 4 and 7 and who have been taking oral corticosteroids for at least 12 weeks prior to the study were enrolled as participants. Participants were treated with a single intravenous injection of either SRP-9001 or a placebo. Primarily, the outcomes analyzed the change in the baseline NSAA score after 48 weeks and the micro-dystrophin expression after 12 weeks. Secondary outcomes assessed the time of the 100 m timed test, time to rise from the floor, and time to ascend four steps, as well as further quantifying micro-dystrophin using IF. After 48 weeks, patients in the treatment group were moved to the placebo group and vice versa, and all patients were treated with intravenous injection once again. The analysis reoccur 48 weeks after this second round of treatment, followed by a 3-year open-label extension for all patients. In January 2021, topline results for the first 48-week period were released [33]. Participants injected with SRP-9001 demonstrated significantly increased micro-dystrophin expression at 12 weeks compared to the baseline (mean expression of 28.1%), and no new safety concerns arose. However, there was no significant improvement of the NSAA score compared to the placebo after 48 weeks. Sarepta suggested that this discrepancy was due to poor randomization, which assigned patients with a higher baseline NSAA to the placebo group, who thus had a better natural history.

In late 2019, Sarepta announced another clinical trial, NCT04626674, known as the ENDEAVOR trial [34]. Thirty-eight patients with the same criteria as in NCT03375164 were confirmed as participants. Interim data from the first 11 patients in this study were released at the World Muscle Society Virtual Congress in September 2021 [35]. These data indicated that the SRP-9001 treatment was associated with robust micro-dystrophin expression localized to the sarcolemma and suggest that the level of expression was related to the vector genome copy number. In addition, adverse effects were consistent with previous studies, and all treatment-related adverse effects were temporary and manageable. Shortly after these data were presented, Sarepta released the results of the integrated analysis of the across studies 101, 102, and 103 at the target dose. At one year, SRP-9001 treated patients improved by 3.1 points (unadjusted means) and 2.4 points (least squared means) on NSAA versus the propensity-weighted external control (*p* ≤ 0.0001) [36]. Therefore, these provide further support to reinforce confidence in our ongoing Phase 3 Study SRP-9001-301, EMBARK, which is a randomized double-blind placebo-controlled trial designed to assess the efficacy of SRP-9001 treatment for patients with DMD [37].

### 3.2. Pfizer-PF-06939926

At the American Society for Gene and Cell Therapy (ASGCT) meeting on May 15, 2020, Pfizer’s press released the preliminary results of its ongoing clinical trial (ClinicalTrials.gov: NCT03362502) [38]. This is an open-label study to primarily evaluate the safety and tolerability of PF-06939926. PF-06939926 is a recombinant AAV9 carrying the mini-dystrophin under the control of a minimized murine muscle creatine kinase (MCK) promoter [39]. The preliminary results were collected from nine DMD patients of 4 years and older taking daily glucocorticoids for at least 3 months, with no pre-existing neutralizing antibodies to AAV9 before the injection of PF-06939926. Three patients received a low dose of 1.0 × 10^14^ vg/kg, and six patients received a high dose of 3.0 × 10^14^ vg/kg. Immunofluorescence (IF) and liquid chromatography mass spectrometry (LCMS) were used to assess the expression of mini-dystrophin. The data indicated that comparing to the normal, 2 months was 20% and 12 months was 24% (*n* = 3). At a high dose, the expression was 35% (*n* = 6) at 2 months and 52% (*n* = 3) at 12 months. Three subjects demonstrated a median improvement of 3.5 points from the baseline in NSAA at 12 months. Fat fraction demonstrated a reduction of 8% at a high dose (*n* = 3) and was unchanged at a low dose at 1 year, estimated by MRI. However, more than 40% patients suffered vomiting, nausea, decreased appetite, and pyrexia. Furthermore, three serious adverse events requiring urgent intervention were observed in the first two weeks following treatment, persistent vomiting, resulting in dehydration, acute kidney injury involving atypical hemolytic uremic syndrome (aHUS)-like with complement activation requiring hemodialysis and eculizumab. All three events were effectively treated and had resolved completely within 2 weeks. Then, the trial was originally put on hold by Pfizer to enable protocol amendments.

During a subsequent update at the Muscular Dystrophy Association’s (MDA) Scientific and Clinical Conference in March 2021, Pfizer announced that it had dosed another 10 patients with PF-06939926, for a total of 19 [40]. Thankfully, there were no more serious AE presented at this extended cohort, which Pfizer attributed to the implementation of a glucocorticoid-based mitigation plan inspired by its previous interim results. A total of 30% of patients in the larger cohort experienced the previously mentioned minor AE, and improvements in NSAA scores were consistent with the previous findings. Based on these promising findings, PF-06939926 received a fast-track designation from the FDA to begin phase-3 trials in late 2020 (NCT04281485) [41], and the first participant was been dosed in in Barcelona, Spain on December 29, 2020. However, in September 2021, three patients were reported to have experienced serious treatment-related muscle weakness, two of whom also suffered from myocarditis, leading to the exclusion of patients who have mutations in exon 9 through 13 or both exon 29 and 30 from all trials [42]. Unfortunately, in December 2021, Pfizer reported the death of a patient being treated with PF-06939926 [43]. No further details have been provided by Pfizer at this point, and both NCT03362502 and the recruitment for the upcoming phase 3 NCT04281485 have been put on hold pending an investigation.

### 3.3. Solid Biosciences-SGT-001

Solid Biosciences initiated a phase 1/2 open-label clinical trial (ClinicalTrials.gov: NCT03368742) designed to assess the safety and preliminary efficacy in DMD patients treated with SGT-001. SGT-001 was the candidate drug using AAV9 and a CK8 muscle-specific promoter to deliver and drive micro-dystrophin carrying the R16/17 neuronal nitric oxide synthase (nNOS) binding domain, which is considered to allow sufficient blood perfusion in the working muscle [44]. Participants in this study are DMD patients who are males between 4 and 17 years, negative for the AAV9 antibodies, and who have used oral corticosteroids for at least 12 weeks prior to beginning the trial. Interim data from the first six patients were made available via a press release from Solid Biosciences in March 2021. Six subjects were treated by SGT-001 with a low dose (5.0 × 10^13^ vg/kg, *n* = 3) and high dose (2.0 × 10^14^ vg/kg, *n* = 3). IF and Western blotting (WB) were used to estimate the expression level of micro-dystrophin. In a patient with the low dose, the micro-dystrophin was below the 5% level of quantification by WB and approximately 10% of fibers by IF. However, in the other two subjects, micro-dystrophin was detected at very minimal levels by IF and none by WB. However, in the three subjects of the high dose, the micro-dystrophin expression ranged from approximately 5–17.5% of the normal dystrophin by WB and 10–70% of positive muscle fibers by IF at day 90. Several days later, the non-ambulatory adolescent DMD patient demonstrated a platelet count reduction, followed by a red blood cell count reduction and transient renal impairment. Patients in the low-dose cohort had a mean NSAA score increase of 1.0, while patients in the high-dose cohort had an increase of 0.3. Patients in the untreated control group experienced a 4-point decline in the same time frame, indicating an initial clinical benefit associated with the treatment with SGT-001. Patients in both dose groups achieved clinical improvements on the 6-min walk test and vital capacity test, as well as meaningful improvements as assessed by patient-reported outcome measures (PROMs). The AEs were serious and the repeated complement activation has resulted in two clinical holds by the FDA. The first was related to thrombocytopenia in 2018 and the second was in October 2019 because of a more widespread complement activation affecting red blood cells (RBCs) and causing renal damage and cardiopulmonary insufficiency [45,46]. Due to the small number of patients tested and the difficulties Solid Biosciences faced in this trial, firm conclusions regarding the safety or preliminary efficacy cannot be drawn from this study at this time. An in vitro study suggests that the AAV capsid not only interacted with various components of the complement system, but also directly activated the complement system in a dose-dependent manner [47].

## 4. Possible Bottlenecks of Clinical Trials of rAAV- Mediated Gene Therapies for DMD

Safety and efficacy are important parameters for gene therapy agents. Due to various advantages, rAAV has become the most widely used delivery vector at present, but it also has its limitations, high-dose AAV will magnify the immune response and increase the safety risk of patients while improving the therapeutic efficiency. For example, clinical data from the AAV8-based gene therapy for X-linked myotubular myopathy (ClinicalTrials.gov: NCT03199469) demonstrated improvement in multiple indicators [48]. However, two years later, three patients treated with the high dose (3 × 10^14^ vg/kg) died of sepsis and gastrointestinal bleeding, successively [49]. Furthermore, Novartis has recently acknowledged that two patients have died of acute liver failure after treatment with its Zolgensma, a one-time gene therapy indicated for some forms of spinal muscular atrophy (SMA) [50]. It is clear that high-dose AAV systemic delivery is an important factor for safety hazards; thus, the development of a dose-dependent relationship between dose and side effects is an inevitable strategy to achieve safe delivery.

Although based on AAV gene therapy to treat DMD broad prospects, it still faces many “bottlenecks” (Figure 2). The curative effects depend on the high-dose AAV, the patient age, the pre-existing immunogenicity against AAV and course of the disease. The three clinical trials of DMD currently recruiting patients with the strong regeneration ability and stable function of muscle cells, no or less pre-existing neutralizing antibodies to the AAV vector, immature immune system. For infants and young children as patients, this stage is regarded as the best “treatment window” of DMD gene therapy. However, few gene therapy drugs have been developed for patients who lose the ability to walk due to inflammation and fibrosis caused by the necrosis of skeletal muscle and heart muscle. Moreover, the above discussed clinical trials of DMD all employed dystrophin as the therapeutic protein, which was treated as a “nonconformist” and induced the immune responses that further exacerbated the adverse events. Here is a more detailed analysis.

### 4.1. Immune Response to High-Dose Systemic AAV May Account for Acute Toxicity

Although the reasons for the apparent difference in the results of the three clinical trials need to be further verified, it is not difficult to see that the DMD patients enrolled in the Sarepta clinical trial are significantly younger (~4.8 years old; immature immune system or small immune response caused by muscle cell necrosis/regeneration), and there is a combination of immunosuppressive agents. Additionally, toxic responses have been researched at various dose ranges in mouse models and large mammals of neuromuscular diseases [51,52,53]. However, no safety concerns arose in mouse models, but when ≥7.5 × 10^13^ vg/kg AAV was delivered intravenously in large mammals, they did arise. In one study [53], when 7.5 × 10^13^ vg/kg vectors were delivered to two 4-year-old rhesus macaques, one developed acute liver toxicity and thrombocytopenia on day 3 and was euthanized on day 5 due to diffuse hemorrhage, the other also demonstrated liver enzyme elevation on day 3. In another study [54], AAV-hu68 (AAV-9 variant) were delivered to three 14-month-old rhesus macaques and three 3- to 30-day-old piglets at the dose of 2 × 10^14^ vg/kg. Of three macaques, one developed acute liver failure and was euthanized on day 5 due to disseminated intravascular coagulation (DIC), the other macaques demonstrated liver enzyme elevation and thrombocytopenia on day 5 and sensory neuron toxicity was observed at the scheduled necropsy on day 28. However, the three piglets did not show liver enzyme, but all demonstrated signs of sensory neuron toxicity within 2 weeks post injection. The researcher suggested that the acute onset of the toxicity might account for the activation of the innate immune responses, which were viewed as an important concern for high-dose systemic AAV gene therapy [47,55,56].

### 4.2. The Mini/Micro-Dystrophin Was Recognized as “Nonconformist” and Destroyed by Overactive Immune Cells, Reducing the Therapeutic Effect

A previous study supports this fact that antibodies against dystrophin were detected as early as 10 days after the inoculation of *mdx* mice with AV expressing dystrophin [57]. Recently, studies have used the German shorthaired pointer (GSHPMD) deletional-null canine model to compare the immune response of dystrophin and utrophin [58]. AAV9 viral vectors containing utrophin and dystrophin were delivered in a single intramuscular injection, respectively. After 4 weeks, it was demonstrated that the muscle tissue on the side receiving mini-dystrophin was almost completely necrotic due to the strong immune response and invasion of a large number of inflammatory cells. The other side of the muscle tissue receiving mini-utrophin did not cause any immune response, and the degree of myocyte necrosis was significantly relieved. The study also pointed out that the dystrophin will be treated as “nonconformist” inducing the immune responses, which will further exacerbate the adverse events.

### 4.3. Fibrosis of Skeletal Muscle Cells and Myocardial Cells Directly Affects the Therapeutic Microenvironment

As mentioned above, comparing to SGT-001 and PF-06939926, SRP-9001 have obtained better therapeutic effect with minimal AEs. The reason is partly attributed to the relatively younger age of the participants that present the mild symptoms and less fibrosis. Furthermore, according to the releases of Pfizer, the deceased patient being treated with PF-06939926 was non-ambulatory. Together, the age of the participants and the severity of the DMD symptoms highly affected the therapeutic effect of the AAV-mediated gene therapy for DMD. Fibrosis is a prominent pathological feature of skeletal muscle in patients with DMD. The deficiency of dystrophin leads to increased sarcolemma permeability, influx of calcium into the sarcoplasm, and activation of proteases leading to necrosis and degeneration of muscle fibers. This, in turn, triggers an inflammatory response to damage repair which leads to chronic inflammation with the persistent production of profibrotic cytokines and excessive synthesis and deposition of fibrosis [59]. A longitudinal study of 25 DMD patients followed for an average of 10 years demonstrated that among the pathological features, only endomysium fibrosis on the initial muscle biopsies correlated with the poor motor outcome gauged by muscle strength and age at a loss of ambulation [60]. This finding supports the notion that fibrosis directly contributes to the progressive muscle dysfunction and the notion that the older the age of the DMD patient, the higher the level of fibrosis. Recently, a study found that following the direct muscle injection of rAAV carrying the expression cassette of γ- sarcoglycan, significant numbers of muscle fibers expressing γ-sarcoglycan and an overall improvement of the histologic pattern of dystrophy were detected in the injected γ-sg-/- mice (the skeletal muscles of mice lacking γ-sarcoglycan). However, these results could be achieved only if injections into the muscle were prior to the development of significant fibrosis in the muscle [61]. Therefore, muscle fibrosis has been recognized as a barrier for muscle gene and stem cell delivery and engraftment.

## 5. Several Strategies Are Currently under Development to Overcome the Bottlenecks

### 5.1. MyoAAV, Which Targets Striated Muscle Transduction, Presents Great Therapeutic Potential for DMD after Injection of a Low Dose of Virus

AAV is one of the most commonly used vectors for in vivo gene replacement therapy and gene editing in preclinical and clinical studies, but the selective delivery of specific tissues after systemic delivery remains a challenge. Recombinant AAV generated by natural capsids is mainly retained in the liver after systemic injection, which greatly limits skeletal muscle and cardiac muscle delivery efficiency. To achieve significant therapeutic effects, an extremely high AAV dose (~2 × 10^14^ vg/kg) is required, which increases the difficulty of vector preparation and quality control. At the same time, a high dose also increases the immunogenicity, as liver injury and other side effects have been observed in some recent clinical trials, increasing the safety risk of AAV gene therapy. In 2020, Weinmann et al. first reported the identification of AAVMYO (muscle-specific capsid variants) containing arginine-glycine-aspartic acid (RGD) motifs, with increased specificity in the murine skeletal muscle, diaphragm, and heart, concurrent with liver detargeting. Moreover, this advance led to the introduction of AAVMYO2 and AAVMYO3 two years later. In addition to retaining all the assets of AAVMYO, AAVMYO2 and AAVMYO3 exhibit a further substantial reduction of off-targeting after the peripheral intravenous delivery [62,63]. In 2021, Tabebordbar et al. further optimized the RGD motif through directed evolution, specific MyoAAVs were selected and validated in mice and non-human primates for their ability to efficiently target the striated muscle transduction and reduce AAV retention in the liver [64]. MyoAAV 2A, one of the AAV variants containing RGD motifs (GPGRGDQTTL) located between Q588 and Q596 of the *cap* genes of the AAV9 screened in this research, demonstrates great therapeutic potential after the injection of a low dose of the virus. As mentioned above, Sarepta-SRP-9001 demonstrated that the administering of a high AAV dose of 2 × 10^14^ vg/kg resulted in the transgene expression in the muscle and functional improvement in the disease phenotype in three clinical trials (clinicaltrials.gov identifiers: NCT03362502, NCT03368742, and NCT03769116). However, in this article, MyoAAV 2A performed high striated muscle transduction efficiency with the low AAV dose in two independent experiments. The in vivo imaging experiment was applied in adult BABL/cJ mice, when injected with AAVrh74, AAV9, or MyoAAV 2A encoding the CMV-Fluc reporter gene with a low dose (2 × 10^11^ vg, representing ~8 × 10^12^ vg/kg). MyoAAV 2A-injected mice demonstrated a dramatically higher bioluminescence signal in the limbs and throughout the body as compared to mice receiving AAVrh74 or AAV9. Furthermore, the systemic delivery of MyoAAV 2A or AAV9 carrying a micro-dystrophin transgene (CK8-microdystrophin-FLAG) into the DBA/2J-*mdx* mouse model of DMD using a low dose (2 × 10^13^ vg/kg) of each AAV was also carried out to estimate the therapeutic efficiency, a quantitative RT-PCR indicated 7.6–15 times higher levels of micro-dystrophin mRNA in skeletal muscles of mice injected with MyoAAV 2A as compared to AAV9. Moreover, MyoAAV 2A delivered 12–46 times higher numbers of vg/dg in skeletal muscles of DBA/2J-*mdx* mice and 2.5 times lower vg/dg in the liver, as compared to AAV9. Surprisingly, while the amount of vg/dg was more than 40 times greater in the liver than in the muscle of AAV9-injected animals, the levels of vg/dg were similar in the liver and muscle of the MyoAAV 2A-injected mice. Furthermore, tibialis anterior muscles of MyoAAV 2A-injected mice recovered significantly greater specific force and were more protected from damage compared to muscles from mice receiving equal doses of AAV9. The research also identified the most potent variants (MyoAAV 4A, MyoAAV 4E, MyoAAV 3A, and MyoAAV 4C) in macaques with higher transduction efficiency for skeletal muscles and de-targeted from the liver when compared to AAVrh74 and AAV9. Finally, the mechanism of the high transduction uncovered the variants’ transduction dependency on the affinity for αV-containing integrin heterodimers, as well as their dependence on AAVR for transduction. Together, these two studies provide important support for each other’s findings, and both studies also provide a theoretical basis for MyoAAV as the potent gene therapy vector for DMD.

### 5.2. Expression Cassette Structure Optimization

#### 5.2.1. Utrophin as the Expression Cassette to Circumvent the Potential Immune Response of Dystrophin

Utrophin is the homologous protein of dystrophin [65], which plays the same function as dystrophin in the embryonic stage [66,67]. Importantly, a normal thymic expression of utrophin in DMD patients could protect utrophin by a central immunologic tolerance [68]. In DMD patients or muscle cell injury, utrophin will be upregulated to a certain extent and redistributed under the muscle cell membrane outside the synapse, partially compensating for the dystrophin function [69,70,71]. Although this self-compensatory upregulation is far from sufficient to improve clinical symptoms, it provides a new strategy for the treatment of DMD.

As an existing protein in DMD patients, transgenic utrophin can avoid an immune response caused by dystrophin delivery. Gilbert et al. firstly used adenovirus (AdV) to deliver a shortened utrophin modeled on the Becker gene [72] and examined the fact that the utrophin was expressed on the sarcolemma and corrected the plasma membrane location of DGC [73]. This result confirmed that the utrophin treatment of DMD is a feasible and effective gene therapy strategy. In 2008, Odom et al. intravenously administered rAAV2/6 harboring a murine codon-optimized micro-utrophin (ΔR4–R21/ΔCT) transgene to five-month-old dystrophin−/−/utrophin−/− (*mdx*:utrn−/−) double-knockout mice [74]. The adult mice demonstrated a localization of micro-utrophin and restoration of DGC to the sarcolemma in all the muscle tested and thereby improved the physiological performance. This type of micro-utrophin also attenuates the muscular dystrophy phenotype in golden retrievers with canine muscular dystrophy (CXMD) [75]. Recently, Song et al. further optimized the cDNA sequence of ΔR4-R21/ΔC, and used AAV9 as the vector to deliver micro-utrophin to 7-day-old *mdx* mice and 7-week-old golden retriever muscular dystrophy (GRMD) dogs [58]. After a few weeks, the striated muscle of the two model animals, especially the myocardium and diaphragm with the earliest and most severe necrosis [76], expressed high levels of mini-utrophin and recruited the missing DGC to help the establishment and function of muscle cells. In addition, after treatment, the number of “central nucleus” muscle fibers, creatine kinase level and muscle capacity were not statistically different from the wild-type, and the symptoms of muscular dystrophy were almost completely cured. This approach may hold promise as a treatment option for DMD because it avoids the potential immune responses that are associated with the delivery of exogenous dystrophin.

#### 5.2.2. Small Activation RNA Targeting the Promoter Region of Mini/Micro-Utrophin Can Be Used to Improve the Transcription Level of Mini/Micro-Utrophin

Small activating RNA (saRNA) is a double-stranded RNA with a length of 21 nt, which can target the promoter region of genes and activate the expression of target genes at the transcriptional level [77]. SaRNA targeting the promoter region of specific genes forms an active complex with the Argonaute protein (Ago) in the cell and is guided into the nucleus. With the help of the Ago complex, the endogenous transcription complex is recruited to the target gene, which promotes the increase in mRNA expression and the upregulation of target protein. At present, saRNA has been widely studied in tumor treatment. Kang et al. used lipid nanoparticles (LNPs) loaded with P21-saRNA and delivered it to bladder cancer model mice by orthopedics injection. They found that P21-saRNA significantly up-regulated the expression of p21 in orthopedic bladder cancer tissues, promoted 40% tumor reduction or disappeared, and significantly prolonged the survival of mice with bladder cancer in situ [78]. P53 is one of the most important tumor suppressor genes in cells. Wang et al. found that p53-saRNA can specifically up-regulate the expression of p53 in tumor cells and effectively inhibit tumor growth and migration [79]. Global first MTL—CEBPA saRNA drug clinical research (ClinicalTrials. Gov: NCT04105335) was approved in 2016. The drug is currently in phase I/II clinical trials. MTL—CEBPA aims to restore C/EBP-α protein to normal levels through the RNA activation mechanism and reduces the immunosuppressive effect of the myeloid cells. In 36 liver cancer patient studies, the MTL-CEBPA has been demonstrated to improve the antitumor activity of cancer therapy by targeting dysregulated myeloid cells and reducing their inhibitory effect in the tumor microenvironment [80]. Based on the above studies, saRNA targeting the promoter region of mini/micro-utrophin was delivered to improve the transcription level of utrophin when constructing mini/micro-utrophin expression elements. This strategy can effectively reduce the dependence of the treatment effect on virus dose.

#### 5.2.3. The 5′UTR Region Was Optimized to Enhance the Translation Level of Mini/Micro-Utrophin

Gene therapy for DMD relies on the construction of the expression cassette structure. A complete gene expression cassette consists of several elements, the promoter (including the enhancer), the 5′-noncoding region (5′UTR), the protein coding region, the 3′UTR and the polyadenine (PolyA) signal. In addition to the selection of the tissue-specific promoter regions, and saRNAs, which were used to improve the transcription level, optimizing the 5′UTR region can improve the expression of core genes at the translational level on account of the recruitment of ribosomes [81]. For example, adding the Kozak motif can improve the level of the ribosome recruitment [82,83]. In recent years, after identifying the 5′UTR sequences that occur naturally in various human cell types with different translational activities, Cao et al. applied genetic algorithms to synthesize 5′UTR libraries. Through a series of screens, they found that adding a specific sequence of 100 bp upstream of Kozak could further improve the translation level [84]. A total of three motifs with the most obvious effect were selected, which were named NeoUTR1, NeoUTR2, and NeoUTR3. In human rhabdomyosarcoma (RD) cells, three 5′UTR motifs each increased the expression of GFP in RD cells by about 34%. The effect of the combined motifs in RD and C2C12 cells was more obvious and demonstrated a combinatorial advantage (CoNeoUTR1-2 > CoNeoUTR 2-1, CoNeoUTR 1-3 > CoNeoUTR 3-1, and CoNeoUTR 2-3 > CoNeoUTR 3-2). For gene therapy of DMD, increasing the 100–200 bp 5′UTR motif can increase the expression of mini/micro-utrophin by enhancing the translation level on the basis of the improved transcription level of saRNA, which provides a powerful means to balance the safe dose with significant efficacy.

### 5.3. CDCs Therapy Modulates Immune Response and Anti-Fibrosis, Providing an Effective Therapeutic Microenvironment for Gene Therapy of DMD

Allogeneic cardiosphere-derived cells (CDCs) are mesenchymal cells derived from atrial or ventricular tissue, so named because they can be cultured in suspension to form a 3D globular cell mass [85].

Clinical studies have demonstrated that allogeneic CDCs play immune modulatory, anti-fibrotic and regenerative roles in DMD and heart failure and are well tolerated in severely affected patients with DMD, but infusion-related hypersensitivity reactions may occur when allogeneic CDCs were sequential intravenous infusions [86,87,88]. The intravenous delivery of CDCs into *mdx* mice can reduce proinflammatory cytokines and alter the expression of genes related to oxidative stress, inflammation, mitochondrial integrity and muscle regeneration by secreting exosomes containing various non-coding RNA and active substances such as mir-148a. Exosomes secreted by CDCs also reprogram fibroblasts and thus have anti-fibrotic effects. More importantly, this phenomenon occurs not only in cardiomyocytes, but also in skeletal muscle cells [89,90]. This means that CDCs extend the “window period” of gene therapy for DMD based on systemic administration, which can greatly help improve the transfection efficiency of gene therapy and benefit more DMD patients.

### 5.4. Immune Suppression May Be the Promising Approach to Manage Capsid or, Potentially, Transgene Immunogenicity

As discuss above, given that the pre-existing neutralizing antibody (NAb) can effectively block rAAV transduction even at low levels and thereby seriously influence long-term transgene expression, patients were excluded if they harbored anti-AAV antibodies in the DMD clinical trials sponsored by Sarepta Therapeutics, Pfizer, and Solid Biosciences, which limited the DMD patients suitable for rAAV therapy. Noteworthily, in clinical and preclinical studies, corticosteroids and other immunomodulatory regimens have been adopted to manage capsid or, potentially, transgene immunogenicity, and some have obtained positive results [91]. In the three DMD trials, large doses of AAV vectors and prophylactic corticosteroids were all administrated, but complement activation relative acute toxicities were only experienced in clinical trials sponsored by Solid Biosciences and Pfizer, but not Sarepta Therapeutics [92]. Furthermore, results obtained from a clinical phase II trial for X-linked myotubular myopathy delivered intravenous rAAV8.AT132 (NCT03199469) of a high dosage group, with 16-weeks of prednisolone commencing 1 day prior to dosing, demonstrating that short-course corticosteroids alone are likely to be insufficient to inhibit the formation of capsid-reactive T cells and rAAV-mediated immune response with systemic high dosages [93,94]. Together, it is possible that these approaches can benefit the DMD treatment and extend the “window period” of patients, but the patients’ characteristics and immune suppression must be thoroughly evaluated to optimize the safe delivery of the high dose systemic rAAV.

## 6. Conclusions

In summary, how to improve and maintain the expression level of utrophin, enhance the targeting of AAV into muscle cells, and reduce liver toxicity are the key issues facing DMD before the clinical trial stage. Undoubtedly, the application of the revolutionary breakthrough of MyoAAV, which specifically transduce the striated muscle combined with the non-immunogenic utrophin, can significantly uncouple the dependence relationship of the therapeutic effect and high dose of AAV. In addition, optimizing the expression cassette with CoNeoUTR, which recruited ribosomes at a high level, and delivering the saRNA, which efficiently and specifically improves the transcription level of the target genes, are the key strategies to promote the safe and effective DMD gene therapy into transformation. Furthermore, CDCs’ cell therapy with anti-inflammatory and anti-fibrosis functions combined with immune suppression can make the delivery of the micro-environment more favorable for MyoAAV, and then more patients can benefit from gene therapy (Figure 3).

Schematic diagram of immune suppression (shown in black arrows) consists of one step. ① Immune suppression in conjunction, pre- or/and post-administration of MyoAAV-mediated gene therapies can manage capsid or, potentially, transgene immunogenicity.

Schematic diagram of CDCs (shown in gray arrows) consists of two steps. ① Cultivation of CDCs. ② System delivery of the CDCs to DMD patients to modulate the immune response and anti-fibrosis, providing an effective therapeutic microenvironment for gene therapy.

Schematic diagram of MyoAAV carrying an optimized mini/micro-utrophin expression. The cassette (indicated in blue arrows) includes eight steps. ① Recognization and internalization. MyoAAV is recognized by the combination of RGD motifs in the capsid of MyoAAV and integrin heterodimer and AAVR of the striated muscle. This triggers the internalization of the MyoAAV via clathrin-mediated endocytosis. ② Transportation. MyoAAV then traffics through the cytosol mediated by the cytoskeletal network. ③ Endosomal escape. Owing to the somewhat low pH environment of the endosome, the VP1/VP2 region undergoes a conformational change and then triggers endosomal escape. ④ MyoAAV undergoes transport into the nucleus and uncoating. ⑤ Transcription. After the single-stranded converted to double-stranded DNA via host cell DNA polymerases or by strand annealing of the sense and antisense strands that may coexist in the nucleus; then, the transcription begins. The transcripts of the mini/micro-utrophin and saRNA were produced under the control of the muscle specific promoter and U6 promoter, respectively. ⑥ Activation of the mini/micro-utrophin transcription. saRNA forms an active complex with Ago 2 and drives the complex to the location of the employed muscle specific promoter and activates the transcription of the mini/micro-utrophin by recruiting transcription factors. ⑦ Nuclear exportion and translation. The transcripts of mini/micro-utrophin export to the cytoplasm and begin translation. The CoNeoUTR sequence can increase the translation by recruiting more ribosomes.

## Figures and Tables

**Figure 1 genes-13-02021-f001:**
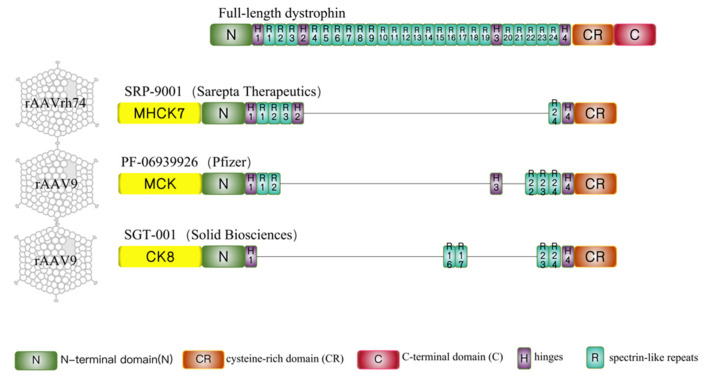
Full-length dystrophin and micro-dystrophins designed in 3 clinical trials. Full-length dystrophin contains an N-terminal domain (N), 24 spectrin-like repeats (R1 to R24), four hinges (H1 to H4), a cysteine-rich domain (CR), and a C-terminal domain (C). SRP-9001(ΔR4-R23/ΔC) is the candidate drug developed by Sarepta Therapeutics and regulated by the MHCK7 promoter. PF-06939926(ΔR3-R21 + H3/ΔC) is used in clinical trials sponsored by Pfizer and driven by the MCK promoter. SGT-001 (ΔR1-R22 + R16R17/ΔC) is a type of micro-dystrophin designed by Solid Biosciences and regulated by the CK8 promoter. The common characteristics of these three micro-dystrophins are the preservation of the N-terminal domain, cysteine-rich domain, and portion of spectrin-like repeats and hinges and the absence of C-terminal domain. The differences are in the central hinges and the R16/17 nNOS-binding domain. SRP-9001 contains hinge 2 and PF-06939926 contains hinge 3, only SGT-001 carries the R16/17 nNOS-binding domain.

**Figure 2 genes-13-02021-f002:**
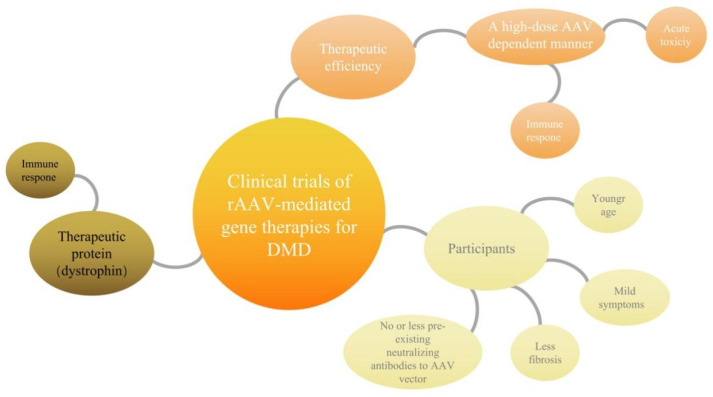
Possible bottlenecks of clinical trials of rAAV-mediated gene therapies for DMD. The possible bottlenecks of the rAAV-mediated mini/micro-dystrophin gene therapy can be summarized into three types. One is that the therapeutic effects are highly dependent on the high dose of rAAV; however, the immune response to high dose systemic rAAV may trigger acute toxicity. The second is that the participants are limited, the three clinical trials of DMD are currently only recruiting infants and young children who have strong muscle cell regeneration ability, stable function of muscle cells, no or less pre-existing neutralizing antibodies to rAAV vector, and the immune system must not yet be mature, which limits the DMD patients suitable for rAAV therapy. The third is that the clinical trials all employed dystrophin as the therapeutic protein, which will be treated as “nonconformist” and induce the immune responses, which will further exacerbate the adverse events.

**Figure 3 genes-13-02021-f003:**
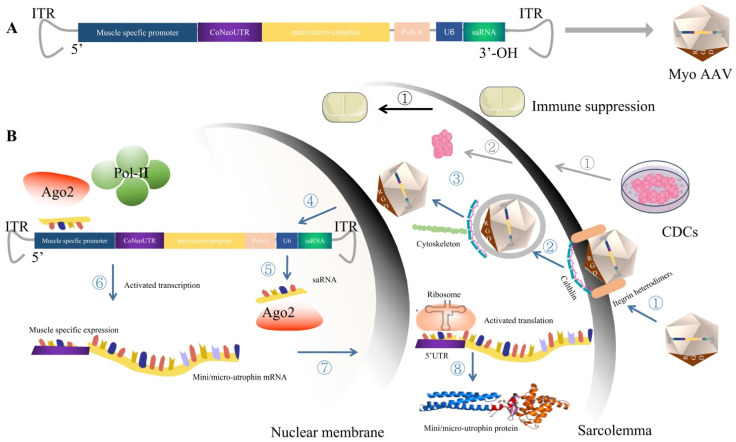
Schematic diagram of CDCs, immune suppression and MyoAAV carrying optimized mini/micro-utrophin expression cassette. (**A**) Overview of the MyoAAV carrying an optimized mini/micro-utrophin expression cassette. The mini/micro-utrophin expression cassette use the muscle specific promoter to regulate the codon-optimized mini/micro-utrophin, and use a U6 promoter to control the rational designed saRNA, which could form an active complex with Ago 2 and driven the complex to the location of the employed muscle specific promoter and activate the transcription of mini/micro-utrophin by recruiting the transcription factors. In addition, CoNeoUTR was added after the muscle specific promoter sequence to activate the translation of the mini/micro-utrophin. (**B**) Schematic diagram of CDCs, immune suppression and MyoAAV carrying the optimized mini/micro-utrophin expression cassette.

**Table 1 genes-13-02021-t001:** Current status of systemic AAV mini/micro-dystrophin clinical trials.

	Sarepta	Pfizer	Therapeutics Solid Biosciences
Trial name	An open-label, systemic gene delivery study using commercial process material to evaluate the safety of and expression from SRP-9001 in subjects with Duchenne muscular dystrophy (ENDEAVOR)	A phase 1b multicenter, open-label, single ascending dose study to evaluate the safety and tolerability of pf-06939926 in ambulatory and non-ambulatory subjects with Duchenne muscular dystrophy	A randomized, controlled, open-label, single-ascending dose, phase I/II study to investigate the safety and tolerability, and efficacy of intravenous SGT-001 in male adolescents and children with Duchenne muscular dystrophy
ClinicalTrials.gov Identififier	NCT04626674	NCT03362502	NCT03368742
Study nature	Phase-1b trial	Phase 1b, open-label, trial	Phase 1/2, open-label, trial
Drug name	SRP-9001	PF-06939926	SGT-001
AAV-serotype	rAAV-rh74	rAAV9	rAAV9
Dose	1 dose (1.33 × 10^14^ vg/kg) for cohort 1	2 doses (1.0 × 10^14^ vg/kg, 3.0 × 10^14^ vg/kg)	2 doses (5.0 × 10^13^ vg/kg 2.0 × 10^14^ vg/kg)
Patient number	38	22	16 (estimated enrollment)
Patient average age	3 years and older	4 years and older	4~17 years
Disease stage	Ambulatory and non-ambulatory subjects	Ambulatory and non-ambulatory subjects	Ambulatory and non-ambulatory subjects
Corticosteroid use	3 months on stable weekly dose of oral corticosteroids for cohort 1	Daily glucocorticoids for at least 3 months	Stable daily dose (or equivalent) of oral corticosteroids ≥ 12 wks
Dystrophin gene mutation	Any mutation	Any mutation	Any mutation
Pre-Nab to AAV	Negative	Negative	Negative
Primary outcome	The change in micro-dystrophin expression in DMD patients treated with SRP-9001.	Safety and tolerability	Safety
Secondary outcome	Adverse events, vector shedding, and the development of antibodies to AAVrh74.	Micro-dystrophin expression in biopsy	
SAE	Increased transaminases that required corticosteroid treatment. Nausea and vomiting that required intravenous treatment (cohort 1).	More than 40% of patients suffered vomiting, nausea, decreased appetite, and pyrexia.	Complement activation, reduced platelet count, liver dysfunction, and acute kidney injury

## Data Availability

Not applicable.

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
