# Peer review of "Strategies for Bottlenecks of rAAV-Mediated Expression in Skeletal and Cardiac Muscle of Duchenne Muscular Dystrophy"

_genes, 2022, doi:10.3390/genes13112021_

Round 1
Reviewer 1 Report
The manuscript by Li and Song describes three clinical trials of rAAV-based therapies for DMD treatment highlighting the current challenges presented by these strategies and commenting about the possible approaches to overcome such challenges. The information provided in the manuscript is overall sufficient to contextualize the current knowledge, however I would suggest that the authors pay more attention to the language and if possible have the manuscript proofread for spelling, syntax and punctuation. To provide an example, at lines 219-220: “The data indicated that comparing to normal, 2 months 20% and 12 months 24% (n = 3).” This sentence appears to be truncated and is not clear in its meaning. Similarly, at lines 294-299: “Although based on AAV gene therapy to treat broad prospects, but its curative effect also depends on the patient age and course of the disease, most of the gene therapy clinical trials currently recruiting for muscle cell regeneration ability strong, stable function of muscle cells, the immune system is not yet mature infants and young children (5 years) for patients with an average age, the stage is regarded as the best "window period" of DMD gene therapy”. Such poorly written sentences are unfortunately common throughout the manuscript and, together with some spelling mistakes, make the manuscript hard to read and in some cases even prevent understanding of the information that the authors would like to convey.
I furthermore have the following minor comments/suggestions for the authors:
1. At line 93, the acronym for Antisense oligonucleotides is ASO or AON as used in the following lines, not ANO.
2. At line 99, the authors mention the drug name “ataluren”, it is not clear to me whether this correspond to the drug referred to as PTC124 at the beginning of the paragraph.
3. At line 104, I think “lease” should be intended as “less”. On this note, at line 389 and in figure 2 legend I also think that “strained” muscle was meant to be “striated” muscle.
4. In figure 1 legend the authors should be consistent in describing the different constructs (deleted domains are currently indicated only for the Sarepta construct). Furthermore, the figure shows that all constructs contain the Cys-rich domain (CR) while the legend affirms that all constructs express the C-terminal domain (C or CT?). The last sentence of the legend wrongly describes the hinges region contained in the SRP-9001 and PF-06939926 constructs (possibly the authors have simply swapped the name of the constructs).
5. When describing the Sarepta-SRP-9001 strategy (section 3.1) could the author describe briefly the MHCK7 promoter as done for the promoters of the constructs used in the other clinical trials described?
6. In the last paragraph of the section reporting results of the Pfizer clinical trial (section 3.2), could the author comment about the potential correlation between specific patient’s mutation and adverse effects?
7. In section 3.3, while describing results from the Solid Biosciences-SGT-001 trial, no expression data information is provided for patients injected with the high viral dose.
8. In section 5.1, could the authors describe what the RGD motifs present on muscle-specific capsid variants are?
9. At line 305 I think the authors meant immature immune system and not autoimmune system.
10. As I am not completely familiar with all the features of dystrophin and DMD, it might benefit also other readers if the authors could comment on why mini-utrophin, unlike mini-dystrophin, is not recognized as “nonconformist” and allows therefore to avoid immune response in patients.
11. In section 5.3 lines 503-507, the sentence is likely truncated and it is therefore not clear the effect of CDCs injection in mdx mice.
12. The legend of figure 2 is not clearly divided in sections A and B like the figure. Also it is impossible to distinguish steps related to CDCs (grey) to steps related to MyoAAV (blue) as the colors are not indicated in the legend.
Reviewer 2 Report
In this review, the authors summarize the latest evidence on the roadblocks of AAV-mediated gene therapy for Duchenne Muscular Dystrophy. Given the clinic trials of DMD treatment, and the need for new avenues for therapy, the discussion of factors challenging DMD therapeutic efficacy is a welcome addition, for specialists and non-specialists alike. The review is well written and appropriately referenced and pitched at a level to give interest an accessibility to a wider audience. Overall, I particularly enjoyed reading the manuscript and feel the authors have made a good contribution. I have following suggestions for authors to consider in revision.
1. The table and figures are welcome addition. I feel this is a little undersold in the manuscript and wondered whether the inclusion of an additional figure to help illustrate what’re the bottlenecks of AAV-mediated gene therapies for DMD.
2. Your article mainly discusses AAV-mediated gene replacement therapy for DMD treatment. It is known that AAV-mediated gene editing (exon skipping, base editing) is emerging strategies for curing DMD. Could you briefly describe the progress of these fields?
3. Does CDCs’ therapy has any side-effects for DMD patients? Could you briefly discuss them in CDCs section?
4. There are typographical problems in this review, such as indentation of the first line. Please make them consistent.
5. Some description should be more rigorous. For example, “As an existing protein in DMD patients that can replace dystrophin, gene therapy delivered by utrophin is the best strategy to avoid immune response caused by dystrophin delivery.” changed into “As an existing protein in DMD patients, transgenic utrophin can avoid immune response caused by dystrophin delivery.” … …
6. It is important to ensure appropriate abbreviations. For instance, “adenovirus” should be abbreviated by “AdV” not “AV”… …
7. Besides MyoAAV, AAVMyo also shows the potent efficacy for transferring skeletal muscle and heart. Please cite the latest literature and describe the main findings of AAVMyo (Jihad El Andari, et al. Sci Adv, 2022 Sep 23;8(38):eabn4704. doi: 10.1126/sciadv.abn4704).
8. For bottlenecks of AAV-mediated gene therapy, the author also needs to mention the pre-existing immunogenicity against AAV in some patients, and how to maintain the long-term effects.
9. Immune suppression has been applied in improving the efficacy of gene therapy in other diseases. Assumedly, it is possible that these approaches can benefit DMD treatment. Thus, the author can briefly discuss them.
10. The manuscript is clear, but there are many grammatical/typographical errors.
10.1 From line 503 to line 507, “In mdx mice that received intravenous delivery of CDCs, which were found to act by secreting exosomes containing various non-coding RNA and active substances such as Mir-148a, which can reduce proinflammatory cytokines and alter the expression of genes related to oxidative stress, inflammation, mitochondrial integrity and muscle regeneration.” was changed into “Intravenous delivery of CDCs into mdx mice can reduce proinflammatory cytokines and alter the expression of genes related to oxidative stress, inflammation, mitochondrial integrity and muscle regeneration by secreting exosomes containing various non-coding RNA and active substances such as Mir-148a”;
10.2 From line 425 to line 428, “Gilbert et al. first tried to use adenovirus (AV) to deliver a shortened version of utrophin modeled on the Becker gene[63]and detected the efficiently expressed utrophin on the plasma membrane of the muscle fibers injected with AV expression utrophin and the expression was sufficient for restore the distribution of DGC in the same site” was changed into “Gilbert et al. firstly used adenovirus (AdV) to deliver a shortened utrophin modeled on the Becker gene[63], and examined that Utrophin was expressed on the sarcolemma and corrected the plasma membrane location of DGC.”
… …
